behaviour, ecology

facultative movements, movement ecology, non-breeding ecology, stable isotopes, tracking, wintering grounds

**Author for correspondence:**
Calandra Q. Stanley
e-mail: callie@umd.edu

# Variable tropical moisture and food availability underlie mixed winter space-use strategies in a migratory songbird

Calandra Q. Stanley[1,2,3], Michele R. Dudash[4], Thomas B. Ryder[1,5], W. Gregory Shriver[6] and Peter P. Marra[1,3]

[1]Migratory Bird Center, Smithsonian Conservation Biology Institute, Washington, DC 20008, USA
[2]Graduate Program in Behaviour, Ecology, Evolution, and Systematics, University of Maryland, College Park, MD 20742, USA
[3]Department of Biology and McCourt School of Public Policy, Georgetown University, 37th and O Streets NW, Washington, DC 20057, USA
[4]Department of Natural Resource Management, South Dakota State University, Brookings, SD 57007, USA
[5]Bird Conservancy of the Rockies, 230 Cherry Street, Suite 150, Fort Collins, CO 80521, USA
[6]Department of Entomology and Wildlife Ecology, University of Delaware, Newark, DE 19716, USA

CQS, 0000-0003-1588-2248

Identifying environmental correlates driving space-use strategies can be critical for predicting population dynamics; however, such information can be difficult to attain for small mobile species such as migratory songbirds. We combined radio-telemetry and high-resolution GPS tracking to examine space-use strategies under different moisture gradients for wood thrush (*Hylocichla mustelina*). We explored the role moisture plays in driving food abundance and, in turn, space-use strategies at a wintering site in Belize across 3 years. Individuals occupying drier habitats experienced lower food abundance and poorer body condition. Using data from our radio-tracked study population and GPS tracking from across five breeding populations, we detected low rates of overwinter site persistence across the wood thrush wintering range. Contrary to expectations, individuals in wetter habitats were more likely to engage in permanent mid-winter relocations, up to 148 km. We suggest facultative movements are instead a condition-dependent strategy that enables wintering wood thrush to locate alternative habitat as food availability declines throughout the dry season. Increased aridity is predicted across the wintering range of wood thrush, and future research should delve deeper into understanding how moisture impacts within and between season space-use dynamics and its ultimate impact on the population dynamics of this declining species.

## 1. Introduction

The role environmental factors (e.g. rainfall, predation, food and their interactions) play in driving the ecology and evolution of most vertebrate species during the non-breeding season is poorly resolved [1]. For small, highly mobile species both logistical and technical constraints have limited our ability to answer even basic ecological questions about many species outside of the breeding season. However, the advent of modern tracking technologies has removed some of these constraints and fuelled an increase in the study of animal movement, and how ecological factors drive movement dynamics and space-use strategies [2,3]. For migratory songbirds, evidence is accumulating of a wide array of both inter- and intra-specific variation in wintering space-use strategies [4,5]. From facultative short-distance shifts in home ranges [6,7] to obligate long-distance intra-tropical movements [8,9], tracking studies are

uncovering a wide range of wintering movement behaviour from ecologically and socially distinct species [10].

Elucidating the environmental factors underlying the diversity of wintering movement strategies for migratory songbirds has been more limited. Food availability is a key ecological factor contributing to winter population limitation [11,12] and winter space-use strategies [6,13]. In wintering American redstarts (*Setophaga ruticilla*), experimental food reductions on individual territories led to diminished physical condition and adoption of a non-territorial strategy by some individuals [14]. At larger spatial scales, tracking studies of Palaearctic migrants on wintering ranges in sub-Saharan Africa have shown that long-distance intra-tropical movements follow shifts in proxies of food abundance (e.g. vegetation indices) [9,15]. Therefore, food availability is likely to be a key factor driving alternative space-use strategies; however, how it interacts with other physiological and environmental conditions to dictate space-use dynamics among individuals, populations and species remains poorly resolved.

In much of the tropics, differential food availability for migratory songbirds has been shown to be driven largely by moisture gradients, primarily through its effect on phytophagous insects [16–18]. The reduction in phytophagous insects during dry periods is of particular relevance as aridity increases towards the end of the dry season (due to seasonal shifts in rainfall), at the same time as birds are physiologically preparing for spring migration. For small migratory warblers, winter rainfall can have important consequences for both spring migration phenology and annual survival [19,20]. Therefore, how migratory birds respond behaviourally and physiologically to increasing aridity and the role aridity plays in shaping space-use strategies could have important implications for understanding the mechanism by which the winter period limits migratory birds.

The aim of this study was to characterize overwinter space-use and determine the underlying correlates of movement in wood thrush (*Hylocichla mustelina*), a long-distance Neotropical migratory songbird. Using radio-tracking data collected across 3 years at a wintering study site in Belize, and remotely from high-resolution GPS-tracking data deployed across five breeding populations, we tested the hypothesis that moisture indirectly drives food abundance, and this in turn shapes space-use strategies. Specifically, we tested the prediction that wood thrush in wetter relative to drier sites will (i) have access to more fruit and insect resources, (ii) maintain better energetic condition and (iii) have higher seasonal site persistence. We combined results from radio-tracking data collected at a wintering study site and remote tracking from high-resolution GPS-tracking data deployed to examine the influence of moisture patterns on movement dynamics.

## 2. Methods

### (a) Study species
Wood thrush are a large (40–50 g), omnivorous Neotropical migratory songbird that breed in eastern North American forests and winter primarily in lowland (up to 500 m) tropical forests of Mexico and Central America. Although a common forest songbird, their global population size has declined by 60% since 1966 based on breeding surveys [21]. During the wintering season, wood thrush are solitary and radio-tracking studies have documented two wintering space-use strategies. Some

individuals maintain flexible territories, while others opportunistically float across territories, a strategy that has been associated with higher mortality [22].

### (b) Wintering field site
Research in Belize was conducted at two adjacent lowland tropical evergreen forest reserves in the late dry season (mid-January through mid-April) from 2013 to 2015. Both forest reserves were located along the Bladen river at the base of the Maya Mountains in the Toledo District of Belize. The Belize Foundation for Research and Environmental Education (BFREE) consists of 467 ha of secondary forest and early successional scrub habitat. The Bladen Nature Reserve consists of 39 270 ha of protected rainforest managed by the Ya'axché Conservation Trust and consists of a mix of secondary and old growth forest in our study area. The area receives an average rainfall of 2500–3000 mm yr$^{-1}$ and experiences a pronounced dry season that lasts from February to April (S. Brewer, BFREE, 1997–2002, unpublished data).

### (c) Site persistence
To monitor seasonal site persistence at our study site in Belize, we equipped a total of 168 birds with radio transmitters (detailed methods in electronic supplementary material, appendix S1). We aged birds by feather characteristics (yearling or adult) and sexed them genetically from blood samples (detailed methods in electronic supplementary material, appendix S1). We radio-tracked a total of 98 adults ($n = 38$ females, $n = 50$ males, $n = 10$ unknown) and 78 yearlings ($n = 32$ females, $n = 37$ males, $n = 9$ unknown). Individuals were tracked until either (i) the bird departed the site or (ii) the transmitter was recovered. Individuals whose transmitters were recovered (e.g. due to mortality, harness failure) were excluded from analyses of seasonal site persistence. Some cases of tag failure or mortality are likely to be included as site departures; however, extensive recapture efforts suggest tag failures were rare.

To assess seasonal site persistence over a larger spatial scale, we examined overwinter location data obtained from adult male wood thrush fitted with archival GPS transmitters (Model PinPoint-50, 1.8 g, 50 fixes, Lotek Wireless) across five breeding populations (Delaware, Indiana, North Carolina, New York and Minnesota; detailed methods in electronic supplementary material, appendix S1). We obtained location estimates from 18 individuals across the winter period, with an average of 11 ± 3.9 location estimates per individual (±s.d.; range 2–17). We defined a winter dispersal event as any movement greater than 1 km after a stationary period (based on two or more locations in the same area) that occurred prior to the commencement of the spring migratory period. Winter home-range size for wood thrush range from 0.1 to 13 h; therefore, movements greater than 1 km should reflect dispersal events [23,24].

### (d) Body condition
We captured birds during two sampling periods (early: mid-January until mid-February; late: mid-March) across the dry season to examine the influence of moisture on body condition. We examined (i) the condition of individuals captured during the early sampling period and (ii) the change in condition of individuals that were recaptured in the late sampling period. Initial body condition was assessed using a body condition index derived by calculating the per cent difference between measured body mass (MBM) and predicted lean body mass (PBM) based on structural size (PBM – MBM/PBM). PBM of each bird was derived based on a regression equation of mass versus wing chord, tail and tarsus for fat-free birds with a pectoral score of 1 [25,26]. Visible fat deposits were scored using a 5-point scale [27] and pectoral muscle was scored following the Gosler scale

[28]. For birds that were successfully recaptured in the late sampling period (mid-March), we examined change in condition by directly measuring change in mass between the early and late sampling period.

## (e) Diet sampling

The influence of moisture on food availability and diet was assessed using (i) fruit and insect sampling, and (ii) isotopic analysis to infer insect consumption. To characterize food availability, fruit and insect surveys were performed at tracking locations randomly selected, without replacement, for individuals that remained at our site for a minimum of two weeks (detailed methods in electronic supplementary material, appendix S1). Surveys were performed in both the early (mid-January until mid-February) and late (mid-March) sampling period to assess change in food availability across the dry season.

To characterize the diet of wood thrush, we quantified the proportion of insects in the diet of wood thrush using stable nitrogen isotopes ($\delta$15N) as an index of food consumption (versus available food resources) from blood and claw samples collected from individuals, regardless of how long they remained at our study site (detailed methods in electronic supplementary material, appendix S1). In general, tissues become more enriched in heavier stable nitrogen isotopes as their diet includes protein from higher trophic level food [29,30]. Variation in turnover rates across tissue types allows the integration of dietary information over different time windows [31,32]. We expect claws to provide an indication of diet integrated over a few months (half-life = 35 days), while whole blood should provide dietary information from a few weeks prior to capture (half-life = 11–16 days) [30,33].

## (f) Patterns of moisture

Patterns of moisture were quantified using (i) an index of moisture derived from remote sensing imagery and (ii) at our study site in Belize using isotopic analysis to characterize habitat moisture. To derive an index of moisture from satellite imagery, we calculated the wetness metric from the tasseled-cap transformation of bi-monthly 30 m resolution Landsat images obtained from our study areas for the period of December–April 2012–2016 (detailed methods in electronic supplementary material, appendix S1) [34]. This metric has been extensively used in tropical landscapes and reflects soil and canopy moisture [35]. We calculated the average and coefficient of variation (CV) of wetness (i) across the dry season and (ii) separately across the early (December–February) and late (March–April) periods of the dry season. At our study site in Belize, we derived the wetness metric across each study plot. Additionally, for individuals that remained at our study site until the late period, we derived the wetness metric across individual home ranges (detailed methods in electronic supplementary material, appendix S1). For GPS-tracked individuals, we derived the wetness metric for each point location estimate.

To obtain individualized indices of habitat moisture (compared with the coarser plot-level measure), we used stable carbon isotopes ($\delta$13C) from blood and claw samples as an index of water availability in the habitats and micro-habitats used by radio-tagged individuals (detailed methods in electronic supplementary material, appendix S1). Differences in stable carbon isotopes are largely driven by photosynthetic pathways in plants (e.g. C3 versus C4 plants). Within forested habitats, water-use efficiency also influences $\delta$13C values within C3 plants, with plants becoming more enriched as water availability decreases [36,37]. Similar to results found in other studies [38,39], we found $\delta$13C values reflected variations in habitat moisture with more enriched values in birds when plots were drier, as measured by satellite-derived wetness indices (claw ($n$ = 147):

estimate = −0.25, s.e. = 0.07, 95% CI = −0.41, −0.11; blood ($n$ = 164): estimate = −0.48, s.e. = 0.16, 95% CI = −0.81, −0.16; more details and results from home-range analysis in electronic supplementary material, appendix S2). As with $\delta$15N (above), difference in turnover rates between claw and blood tissue allows us to examine changes in environmental conditions over different time windows.

## (g) Data analysis

For individuals that remained in our study area until the end of the dry season, we characterized how ecological conditions changed over time. We used paired $t$-tests to compare changes in average home-range wetness and mass for individuals recaptured across sampling periods. To assess change in food availability, we performed separate Wilcoxon signed-rank tests to compare fruit and insect abundance on home ranges.

To determine the influence of moisture levels on food availability (fruit and insect surveys) and diet ($\delta$15N), we constructed separate models with fruit abundance, insect abundance or $\delta$15N as the response variable. We developed a candidate model set for each response variable based on the predictions that moisture will directly influence food availability and diet. We hypothesized that condition at capture and dominance relationships between age and sex classes may influence food availability and diet. For food availability models, we focused analysis on the early sampling period ($n$ = 44 fruit surveys; $n$ = 43 insect surveys) due to low site persistence of radio-tracked wood thrush. Home-range size, home-range-specific estimates of wetness and year were also included as fixed effects in the food availability models. Fruit abundance were log-transformed and fitted with a general linear model (GLM), while insect abundance was modelled with a GLM with a Poisson distribution. For the analysis of diet, we modelled $\delta$15N values from both sample periods with a linear mixed-effect model. Period-specific estimates of plot wetness and habitat moisture ($\delta$13C) were used. Sampling period and year were also included as a fixed effect and bird identity was included as a random effect.

To determine the influence of moisture patterns on body condition, we constructed separate GLMs using (i) the body condition index for all birds captured during the early sampling periods or (ii) weight change across the sampling periods for recaptured individuals as the response variable. For both measures of body condition, we hypothesized that moisture would indirectly influence body condition through its effect on insect availability as measured by $\delta$15N values. Plot wetness and habitat wetness ($\delta$13C) were included to account for other possible mechanisms in which moisture may influence body condition. We included sex and age classes to explore the influence of dominance relationships. Year was also included as a factor, and for the analysis of weight change, time since capture was included.

We modelled seasonal site persistence (time to event outcome) using Cox proportional-hazards (CPH) regression models [40,41]. CPH models allow estimation of the effect of covariates on the hazard of departure (hazard ratio), or risk of departure. We developed a set of *a priori* models to examine the influence of moisture on seasonal site persistence for radio-tracked birds in Belize. Similar to models above, we examined the indirect impact of moisture on insect availability as measured by $\delta$15N values. We included plot wetness and habitat moisture ($\delta$13C values) as direct measures of moisture, to account for other mechanisms driven by moisture levels. We also included condition at capture, age and sex class to explore the influence of dominance hierarchies [42]. Year was included as a factor. For individuals tracked using GPS tags, we constructed a more limited candidate set of models that included the average and the CV of wetness across location estimates and winter latitude. For individuals that performed intra-winter movements, we used paired $t$-tests to compare the average and the CV of wetness between location estimates pre- and post-movement.

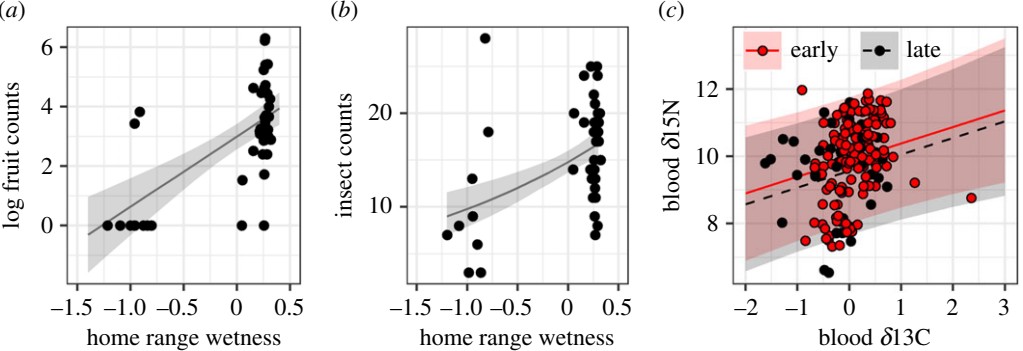

**Figure 1.** Factors driving (*a*,*b*) food availability and (*c*) food consumption for wood thrush during the wintering season. (*a*) Fruit and (*b*) insect surveys conducted on individual wood thrush home ranges were positively related to satellite-derived home-range wetness. Fruit counts were log-transformed. Satellite-derived home-range wetness was scaled, more positive wetness values denote wetter habitats. (*c*) Positive relationship between stable nitrogen ($\delta$15N) and stable carbon ($\delta$13C) values derived from blood during the early (red, solid line) and late (black, dashed) sampling period. More positive $\delta$15N values denote larger proportions of insects in diet. More positive $\delta$13C values denote drier conditions. Lines denote the model predicted relationship and shaded area the 95% CI. (Online version in colour.)

For all models including stable isotopes ($\delta$15N or $\delta$13C), separate models were run for isotopes derived from blood or claws. All analyses were conducted in R 3.4.3 [43]. Linear and generalized mixed-effect models were run in the R package *lme4* with restricted maximum-likelihood estimation [44]. GLMs were run in the R package *stats* [43]. CPH models were performed using the *survival* package in R [45]. To facilitate interpretation of the relative strength of parameter estimates, all input variables were standardized by subtracting the mean and dividing by two standard deviations following Gelman [46,47]. Therefore, parameter estimates provided represent standardized effect sizes that are on a comparable scale. Parameter estimates with confidence intervals that exclude zero were interpreted as being important predictors in a model. For full model specifications and fit assessment, see electronic supplementary material, appendix S1.

## 3. Results

The majority of radio-tracked wood thrush (68%; $n = 115$) departed our study sites in southern Belize prior to the start of the spring migratory period (date range: 25 January–26 April; electronic supplementary material, appendix S1 and table S1). Most individuals ($n = 44$) occupied a contiguous forested area moving an average of $55.8 \pm 89.20$ m between relocations. However, a number of individuals ($n = 9$) made regular off-territory forays (up to 2 km away). As the dry season progressed, there were significant changes in environmental conditions on our study sites. In general, home-range wetness ($n = 75$, $z = -2.01$, $p = 0.045$) and insect abundance declined ($n = 75$, $z = -2.51$, $p = 0.013$) while fruit abundance increased between our two sampling periods ($n = 78$, $z = -3.30$, $p < 0.001$; electronic supplementary material, appendix S1, figure S1). Home-range wetness was significantly lower in the 2013 season compared with 2014–2015 ($F_{2,49} = 1270$, $p < 0.001$, $n = 52$; Bonferroni adjustment: 2013–2014, $p < 0.001$; 2013–2015, $p < 0.001$; 2014–2015, $p = 1$; electronic supplementary material, appendix S1, table S1) and corresponded to overall lower dry season rainfall across Belize [48].

### (a) Food availability and diet

Home-range wetness was included in the final model explaining both fruit and insect abundance (figure 1 and table 1; electronic supplementary material, appendix S1, table S2).

**Table 1.** Model-averaged parameter estimates, standard error and unconditional 95% confidence intervals for top linear mixed-effect models analysing factors driving variation in food availability (fruit abundance and insect abundance) and food consumption (proportion of insects consumed, $\delta$15N) for non-breeding wood thrush in Belize. Food consumption models included isotope values derived from blood which provides information on diet and habitat use over a few weeks. Models were ranked by comparing Akaike's information criterion values corrected for small sample (AIC$_c$). All models with $\Delta$AIC$_c \le 2$ were considered equally plausible. Values italicized have confidence intervals that do not include zero.

| parameter | estimate | s.e. | 95% CI |
|---|---|---|---|
| fruit abundance (*n* = 44) | | | |
| *home-range wetness* | *2.36* | *0.44* | *1.50, 3.22* |
| insect abundance (*n* = 43) | | | |
| *home-range wetness* | *0.40* | *0.10* | *0.21, 0.59* |
| *home-range area* | *−0.19* | *0.09* | *−0.36, −0.01* |
| age (yearling) | 0.12 | 0.08 | −0.04, 0.29 |
| sex (male) | −0.14 | 0.08 | −0.30, 0.03 |
| food consumption ($\delta$15N, blood; *n* = 165) | | | |
| *habitat moisture ($\delta$13C)* | *0.47* | *0.19* | *0.11, 0.84* |
| *period* | *−0.35* | *0.17* | *−0.68, −0.01* |
| plot wetness | −0.29 | 0.20 | −0.44, 0.26 |

Individuals on wetter home ranges had access to more fruit (figure 1*a* and table 1) and insects (figure 1*b* and table 1). Additionally, the final model describing insect abundance included age, sex and home-range size. Insect abundance was also higher for individuals on smaller home ranges (table 1). The confidence intervals for both age and sex crossed zero (table 1). We also found evidence that the proportion of insects consumed by wood thrush, based on stable nitrogen isotopes, was influenced by habitat moisture ($\delta$13C) and sampling period (table 1; electronic supplementary material, appendix S1, table S3). Similar results were obtained for averaged models constructed with blood versus claw isotopes, here we will focus on results from blood isotopes which provides information on recent diet (results from claw isotopes: electronic supplementary material, appendix S1, tables S3 and S4). The final model indicated higher insect

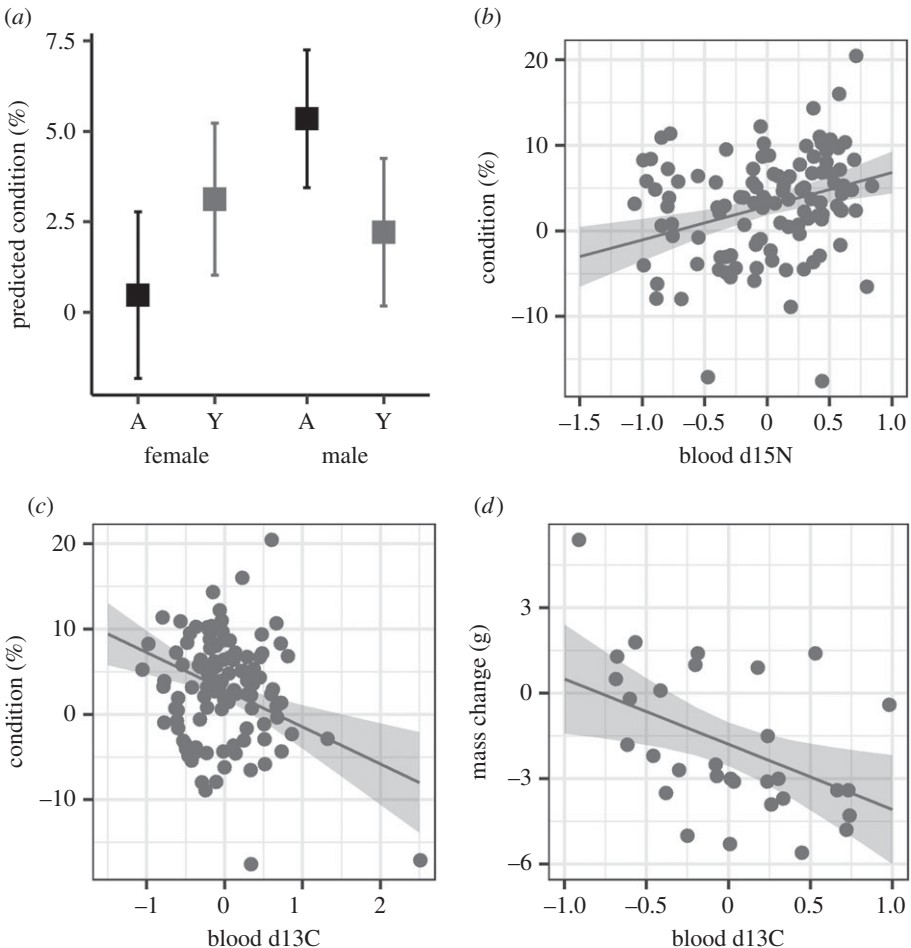

**Figure 2.** Factors driving body condition of radio-tracked wood thrush during the wintering season in Belize. (*a*) Body condition index (marginal mean ±95% CI) during the early dry season by age (A, adult; Y, yearling) and sex class. (*b*) Positive relationship between stable nitrogen values in blood and body condition index. Stable nitrogen values scaled, such that more positive values denote larger proportions of insects in diet. (*c*) Body condition index and (*d*) change in mass were negatively related to stable carbon values in the blood. Stable carbon value have been scaled, more negative values denote drier habitats (note the difference in axis scale). For scatterplots, line denotes the model predicted relationship and shaded area the 95% CI.

consumption (more enriched $\delta$15N) when habitats were drier (more enriched $\delta$13C; figure 1*c* and table 1) and a shift to consuming fewer insects (more depleted $\delta$15N) during the late sampling period (table 1). The final model also included plot wetness but the confidence intervals included zero.

## (b) Body condition

We detected a decline in wood thrush body condition across the late dry season, with individuals losing on average 1.4 ± 2.7 g (±s.d.; per cent body mass: 2.9 ± 5.83% s.d.; less than 2 months; $t = 3.23$, d.f. = 37, $p = 0.0013$). Similar results were obtained for averaged models examining the influence of moisture on body condition that included blood versus claw isotopes (electronic supplementary material, appendix S1, tables S5–S7). Given that blood isotopes represent more recent environmental conditions and therefore better reflect current body condition, we focus on results derived from blood isotopes. During the early sampling period, we found evidence of an effect of habitat moisture, diet and an interaction between age and sex class on body condition (figure 2*a–c* and table 2). Individuals in wet habitats (depleted $\delta$13C values) were in better condition (figure 2*b*). Body condition was also higher in individuals that consumed a higher proportion of insects (enriched $\delta$15N values; figure 2*c*). Finally, the averaged model predicted body condition would be highest in adult

males while lowest in adult females (figure 2*a*). We found evidence that change in body mass was influenced by habitat moisture, plot wetness and time since capture (table 2; electronic supplementary material, appendix S1, tables S6 and S7). Individuals in drier habitats (enriched $\delta$13C signatures) lost more weight over the sampling periods (figure 2*d* and table 2). The confidence intervals for the estimates of plot wetness and time since capture overlapped zero (table 2).

## (c) Seasonal site persistence

At the beginning of the spring migratory period (late April; [26]), the daily probability of seasonal site persistence of radio-tracked birds at our Belize study site was 20.0% (95% CI 0.13–0.31; $n = 23/92$ birds remained on site). The top model analysing site persistence only found evidence of an effect of moisture on the probability of departure (figure 3*a*; electronic supplementary material, appendix S1, table S8 and figure S3). Contrary to expectations, individuals had a higher departure probability when plots were wetter based on the satellite-derived wetness metric (model with isotopes derived from blood ($n = 101$): estimate = 0.64, s.e. = 0.21, 95% CI = 0.22, 1.06; claws ($n = 91$), estimate = 0.61, s.e. = 0.22, 95% CI = 0.19, 1.04; figure 3*a*).

Seasonal site persistence across the wintering range was assessed by examining wood thrush fitted with archival GPS

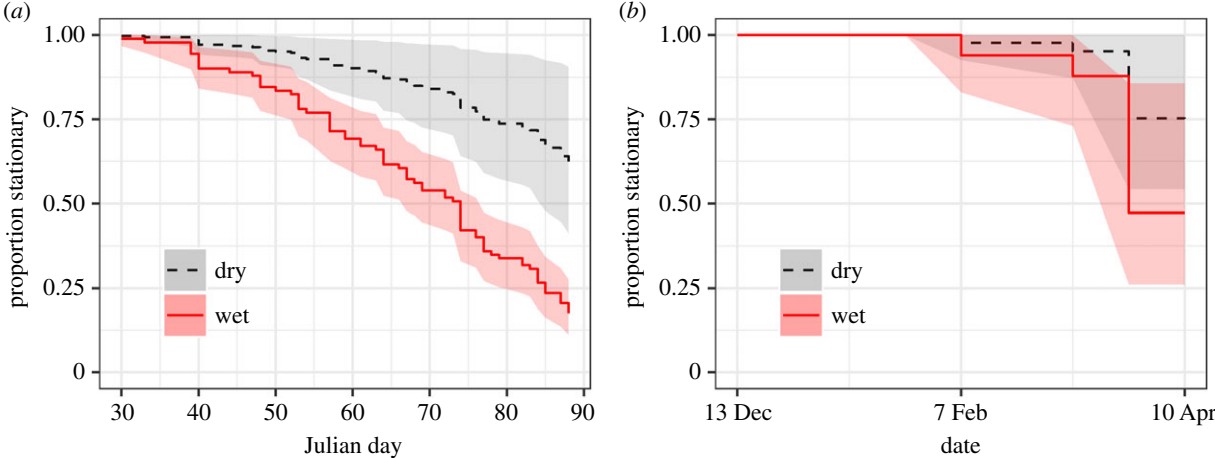

**Figure 3.** Proportion of wood thrush remaining stationary across the wintering period from our radio-tracked study population in Belize and wood thrush tracked using GPS transmitters based on the estimated survival function of a Cox proportional hazard model. In our radio-tracked population (*a*) and GPS-tracked population (*b*), estimated proportion of wood thrush remaining stationary in wet (solid, red line) and dry (dashed, black line) habitats based on satellite-derived wetness values. Dry values are expressed as the first quartile of wetness values; wet values are expressed as the third quartile of wetness values. (*a,b*) Line represents the estimate of proportion of stationary individuals from model, shaded area 95% CI. (Online version in colour.)

**Table 2.** Model-averaged parameter estimates, standard error and unconditional 95% confidence intervals for top linear mixed-effect models analysing factors driving body condition index in the early dry season sampling period ($n = 112$) and change in mass between the early and late dry season sampling period ($n = 31$) for non-breeding wood thrush in Belize. Models included isotope values derived from blood which provides information on diet and habitat use over a few weeks. Models were ranked by comparing Akaike's information criterion values corrected for small sample (AIC$_c$). All models with $\Delta$AIC$_c \leq 2$ were considered equally plausible. Values italicized have confidence intervals that do not include zero.

| parameter | estimate | s.e. | 95% CI |
|---|---|---|---|
| **early body condition index** | | | |
| *habitat moisture (δ13C, blood)* | *−4.35* | *1.19* | *−6.68, −2.03* |
| *diet (δ15N, blood)* | *3.93* | *1.13* | *1.7, 6.15* |
| wetness | −1.10 | 1.13 | −3.33, 1.12 |
| age (yearling) | −0.55 | 1.06 | −2.63, 1.52 |
| sex (male) | 1.98 | 1.08 | −0.14, 4.1 |
| *age (yearling) × sex (male)* | *−5.79* | *2.13* | *−9.96, −1.62* |
| **change in mass** | | | |
| *habitat moisture (δ13C, blood)* | *−2.59* | *0.89* | *−4.33, −0.84* |
| time | −1.60 | 0.88 | −3.33, 0.13 |
| wetness | 1.03 | 1.03 | −0.98, 3.04 |

transmitters on five breeding sites (electronic supplementary material, appendix S1, table S9 and figure S2). At the commencement of the spring migratory period for our GPS-tracked population, the probability of site persistence was higher, 66% (95% CI = 0.44–0.97; $n = 7/17$ individuals) compared to our radio-tracked population. The first GPS-tracked individual to depart its initial home range occurred at the end of February (electronic supplementary material, appendix S1, table S1). The final model analysing site persistence found an effect of moisture on probability of departure ($n = 16$, figure 3*b*; electronic supplementary material, appendix S1, table S10 and figure S3). Similar to wood thrush in Belize

and contrary to expectations, the probability of departure was higher for individuals in wetter habitats (estimate = 1.10, s.e. = 0.48, 95% CI = 0.16, 2.04; figure 3*b*) and habitats with higher variation in wetness (estimate = 0.83, s.e. = 0.4, 95% CI = 0.05, 1.61). On average, individuals that departed initial home ranges moved $48.2 \pm 57.8$ km (range: 1.3–182.3 km; electronic supplementary material, appendix S1, figure S2); whereas individuals that remained site persistent moved an average of $124 \pm 159.5$ m ($\pm$s.d.) between relocations (every two weeks). There was no difference in the average wetness or variability in wetness of locations occupied by individuals pre- or post-movement (average wetness: $t = 0.68$, $p = 0.263$, d.f. = 5; CV wetness: $t = 1.07$, $p = 0.168$, d.f. = 5).

## 4. Discussion

By combining on-the-ground fieldwork on wintering grounds with high-resolution GPS-tracking data deployed on breeding adults, we were able to demonstrate that space-use strategies of wood thrush throughout their wintering range was dependent on patterns of moisture which influenced food availability. For our radio-tracked study population in Belize, individuals in drier habitats had lower food resources and were in poorer body condition. We found that as habitats became more arid across the dry season, insect abundance declined (although fruit abundance increased) and birds in drier habitat experienced larger declines in body condition. Overwinter site persistence was low (0.20, 95% CI 0.13–0.31), and contrary to expectations, departure probability was greater when habitats were wetter and had higher apparent food availability. GPS-tracking of birds from across the breeding range confirmed this pattern of mid-winter movements and revealed that individuals relocated over large distances (range: 1.3–148.0 km), regardless of winter latitude. These results provide evidence that space-use strategies in overwintering wood thrush are flexible, and at least partly driven by changes in environmental conditions linked to patterns of moisture.

Although unexpected, our finding that wood thrushes frequently made permanent, long-distance, mid-winter

relocations using two independent tracking data sources underscores the biological importance of these findings. Prior research on wood thrush winter ecology has shown that wood thrush can employ a floater or 'wanderer' strategy (e.g. expanded home ranges or large movements) [22]. However, these studies were conducted at smaller spatio-temporal scales (transmitter life < 20 days), which makes characterizations of the scale, permanence and relevance of these movement difficult. Although departure probability was high for both study populations, some individuals abstained from movements, suggesting this strategy is not obligate and shows both among and within individual variation. For example, individuals that were radio-tracked across multiple years from our Belize population (n = 17) showed variability in seasonal site persistence across years (e.g. timing, site tenacity). These results suggest that winter movements in wood thrush are a context-dependent facultative strategy that allows for flexibility in space-use among and within individuals. These results not only contribute to growing evidence that migratory birds engage in a wide array of movement behaviours during the wintering season [10], but also elucidate the ecological factors that drive these behaviours.

Mounting evidence points to the importance of food availability as the key limiting factor for migratory songbirds during the winter period and the major driver of individual space-use behaviour [6,11,26]. The results presented here support the conclusion that habitat quality is linked to food availability, which is ultimately driven by patterns of moisture. On drier home ranges, with lower fruit and insect abundance, individuals were more likely to be in poor body condition and lose weight across the dry season. We also found that both females and young birds were more likely to be in lower body condition suggesting behaviourally dominant adult males may exclude subordinate individuals from settling in the highest quality home ranges, as has been found in other neotropical migrants [12,42]. In American redstarts, competitive exclusion by dominant adult males forces subordinate individuals into drier low quality winter habitat which ultimately carry-overs to limit reproductive success the subsequent breeding season [49]. Here, we did not observe segregation across different habitat types; however, the differences in condition we observed suggest habitat quality for wood thrush may also vary across finer spatial scales. Therefore, high-quality home ranges may be limited and less competitive birds are behaviourally excluded to drier, suboptimal home ranges. Wood thrush arrive on their wintering range at the end of the wet season (October–November), when insect abundance is high [50], but as the dry season progresses, insect abundance declines [26]. Therefore, differences in habitat quality are likely to be exacerbated towards the end of the dry season and could have important consequences for individual performance and preparation for their energetically demanding migration back to their breeding grounds.

Facultative movements recorded from individuals tracked with GPS transmitters confirm that this behaviour occurs primarily during the latter part of the winter season (February–April). Alternative space-use strategies are expected to maximize survival under suboptimal conditions, such as when optimal habitats are saturated or food resources are unpredictable [13]. The timing of facultative movements in wintering wood thrush suggests that the ultimate mechanism driving departure may be deteriorating environmental conditions, driven by declines in moisture across the dry season [51]. This behaviour could be akin to the tracking of resources described for long-distant intra-tropical movements [4,9] and altitudinal migrants [52]. For wood thrush, seasonal declines in moisture are more predictable and the scale of movements is smaller, but similarly, birds may be initiating movements in search of improved environmental conditions. The broad range of departure dates across all years suggests that individuals were not responding to broad-scale environmental cues (e.g. photoperiod, phenology) [5,9], but may instead be responding to finer-scale changes in habitat suitability. Therefore, facultative movements in wood thrush could be an adaptive response to fine-scale fluctuations in habitat suitability.

Contrary to our predictions, individuals had a higher probability of engaging in facultative movements when habitats were wetter. Furthermore, neither sex nor age class influenced the probability of departure. This suggests that facultative movements are not driven by competition avoidance by subordinate individuals (e.g. competitive release) [53]. Given the evidence that birds were more likely to engage in winter movements when habitats were wetter and birds were overall in better condition, we propose that facultative winter movements in wood thrush may be condition-dependent [54]. We did not find direct evidence that body condition influenced the probability of departure. However, our measure of body condition may not reflect the conditions experienced by individuals when movement decisions were made due to the temporal mismatch between when birds were first captured and when they departed our site. We propose that, as local habitat suitability declines, individuals experiencing poor conditions may be physiologically constrained from engaging in this putatively risky behaviour (e.g. due to low fat or pectoral muscle) and may instead try to 'ride out' deteriorating conditions. Therefore, when habitats are drier, such as in 2013, birds on average remain on our study plots longer. Given that birds must prepare for spring migration at the end of the dry season, individuals that attempt to 'ride out' conditions may be forced to delay migration preparations should they remain and conditions not improve in time.

While we suggest that facultative wintering movements by wood thrush are ultimately a response to declining habitat suitability, we found no evidence that individual wood thrush were directly tracking seasonal resources [9] or upgrading to higher quality habitats [6]. Based on GPS tracking, there was no difference in the overall average wetness, a good proxy of food abundance, or variability in wetness between locations pre- or post-movements. However, low sample size made it difficult to account for latitudinal and elevational differences among individuals. Therefore, a large question left unanswered is the proximate mechanism driving timing of movement decisions and settlement in new locations. Additional factors that could drive habitat suitability and influence decision-making of individuals that should also be considered include changes in competition [10] or predation risk. For example, if increased moisture also drives increased movement of conspecifics or heterospecifics (e.g. altitudinal migrants [52]), increased competition for resources could drive movement decisions. However, we did not find support for competitive release by subordinate individuals. Although predation rates were relatively low in our radio-tracked population (n = 23/181 individuals), increased density of predators when habitats are wetter could also drive movement decisions

and deserves further exploration [55]. Comparing the fitness consequences (e.g. survival, condition, reproductive success) of either strategy will be necessary to better resolve the adaptive value of facultative winter movements in wood thrush.

Deciphering the role that seasonal patterns of moisture play for limiting populations of Neotropical migratory songbirds has important implications for the management of these species, as climate models forecast increasingly arid condition on their wintering range [56]. In this study, we found that individuals in drier conditions were in lower individual body condition and were less likely to depart wintering home ranges as conditions deteriorated across the dry season, the consequences of which are still unclear. Previous research has suggested that wood thrush that float across home ranges face higher mortality rates ('wanderers' *sensu* Rappole *et al*. [22]). If facultative movements are a behavioural strategy to avoid deteriorating dry season habitat suitability, this strategy may buffer individuals against the predicted increased severity of dry seasons if they can locate suitable habitat. However, if facultative movements are a condition-dependent strategy, as our data suggest, then increasingly dry conditions during the wintering period could prevent birds from escaping deteriorating conditions as the dry season progresses. Developing a better understanding of the mechanisms driving facultative movements of wood thrush, and perhaps other species of migratory birds, will be essential to understanding the flexibility individuals may need to exhibit to adjust to future climatic shifts.

Ethics. This research was approved by the National Zoological Park Institutional Animal Care and Use Committee (Protocol 14-04) and University of Delaware Institutional Animal Care and Use Committee (Protocol AUP-1157). Fieldwork in Belize was carried out under Belize Forest Department Scientific Collection/Research Permits (CD/60/3/13 (07), CD/60/3/14 (11), CD/60/3/15 (9)). Fieldwork in the USA was carried out under US federal banding permits (09700, 21405, 23218, 23475) and Delaware (2016-WSC-032), Indiana (NP15-12, NP16-19), Minnesota (19616), New York (Banding no. 157) and North Carolina (15-SC01020) state research permits.

Data accessibility. The data are provided in electronic supplementary material [57].

Authors' contributions. C.Q.S.: conceptualization, data curation, formal analysis, investigation, methodology, validation, visualization, writing—original draft, writing—review and editing; M.R.D.: conceptualization, resources, supervision, writing—review and editing; T.B.R.: conceptualization, funding acquisition, project administration, resources, supervision, writing—review and editing; W.G.S.: funding acquisition, investigation, project administration, resources, supervision, writing—review and editing; P.P.M.: conceptualization, funding acquisition, project administration, resources, supervision, writing—review and editing.

All authors gave final approval for publication and agreed to be held accountable for the work performed therein.

Competing interests. We declare we have no competing interests

Funding. Funding for this study came from a Department of Defense Strategic Environmental Research & Development Program (RC-2121), Minnesota National Park Service, Bedford Audubon Society, Forsyth Audubon Society, National Audubon Society and a United States Department of Agriculture McIntire-Stennis Capacity grant no. (DEL00702) and the BEES/BISI program.

Acknowledgements. This work would not have been possible without the field assistance of M. Cruz, L. Duval, N. Freeman, J. Hightower, C. Hines, A. Sutton and L. Welsh in Belize. We also thank numerous field assistant at our field sites in the USA, notably T. Guida and P. Keller and the volunteers from Forsyth Audubon, Bedford Audubon, Carpenter Nature Center and Warner Nature Center for assistance at our field sites in the USA. We would like to thank J. Marlin, H. Barrett and J. Dourson for their logistical support at BFREE, and S. Adalsteinsson, K. Brand, Z. Landin, J. Robbins, K. Serno, P. Smithson, K. Thorington and J. Veith for their logistical support in the field in the USA. We would also like to thank H. F. James, J. Song and G. Wilkinson for helpful comments on the manuscript.

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
