## [Peer Review File · Proceedings of the Royal Society B: Biological Sciences]

Review History

RSPB-2020-2608.R0 (Original submission)

Review form: Reviewer 1

Recommendation

Accept with minor revision (please list in comments)

Scientific importance: Is the manuscript an original and important contribution to its field?

Good

General interest: Is the paper of sufficient general interest?

Good

Quality of the paper: Is the overall quality of the paper suitable?

Excellent

Is the length of the paper justified?

Yes

Should the paper be seen by a specialist statistical reviewer?

No

Do you have any concerns about statistical analyses in this paper? If so, please specify them explicitly in your report.

No

It is a condition of publication that authors make their supporting data, code and materials available - either as supplementary material or hosted in an external repository. Please rate, if applicable, the supporting data on the following criteria.

Is it accessible?

Yes

Is it clear?

Yes

Is it adequate?

Yes

Do you have any ethical concerns with this paper?

No

Comments to the Author

This study successfully combines a wide array of novel approaches to identify environmental correlates driving space-use strategies in a Neotropical migratory bird. The manuscript is well written and provides an interesting contribution to this topic. I have several comments to further improve clarity of the paper.

L 39-40: Keywords should not repeat words from the title

L 51, 59, 346: I suggest to use "movements" instead "migration" here, because intra-tropical "migration" typically differs from directional intercontinental migration by the absence of returning to the starting location.

L 81: You probably mean body condition as you do not use the term "energetic condition" hereafter (see also L 297 and 299).

L 103: "from February-April" should be "from February to April"

L 180-181: Here I suggest to take the time span between the capture and the recapture of each individual into account - the longer span may lead to larger magnitudes of the focal changes and vice versa.

L 227, 278-279, 287: Please check the reference errors.

L 300: I suggest to emphasize the contrast with your expectations in the results section as well.

L 326-328: It is interesting that "females and young birds were more likely to be in lower body condition suggesting behaviourally dominant adult males may exclude subordinate individuals from high-quality sites" but in L 354-356 you state that "neither sex nor age class influenced the probability of departure" and "facultative movements are not driven by competition avoidance by subordinate individuals". However the exclusion of the subordinate females from high-quality sites would lead to movements which, based on your claim, should not be related to sex. Please explain better and discuss briefly.

Fig. 2: Please explain in the figure legend what A and Y on x-axis mean.

Table 1: Please shortly explain how the top models were selected (or refer to the respective section for more details).

Appendix S1: "Ctmm" should be "ctmm" in 4th item in the list of references.

Table S1: The order of supplementary tables should be changed (or missing link to Table S1 added), as you refer to other supplementary tables prior to Table S1.

Review form: Reviewer 2

Recommendation

Major revision is needed (please make suggestions in comments)

Scientific importance: Is the manuscript an original and important contribution to its field?

Excellent

General interest: Is the paper of sufficient general interest?

Good

Quality of the paper: Is the overall quality of the paper suitable?

Acceptable

Is the length of the paper justified?

Yes

Should the paper be seen by a specialist statistical reviewer?

No

Do you have any concerns about statistical analyses in this paper? If so, please specify them explicitly in your report.

Yes

It is a condition of publication that authors make their supporting data, code and materials available - either as supplementary material or hosted in an external repository. Please rate, if applicable, the supporting data on the following criteria.

Is it accessible?

Yes

Is it clear?

Yes

Is it adequate?

Yes

Do you have any ethical concerns with this paper?

No

Comments to the Author

GENERAL:

This manuscript presents a most interesting data set on the winter strategies of a long-distance migratory songbird. The combination of local-scale and large-scale tracking makes the data ideal for investigating both local individual and larger-scale continental factors. It nicely shows correlations between food and wetness, better condition of birds in wet habitat and it demonstrates surprising higher likelihood of relocation in birds from wetter habitats. However, I found the methods and results section quite challenging to understand and thus, I might easily have misunderstood (and missed) parts. Furthermore, I feel that the difference between dry and wet years and the distinction between condition/bird quality and wetness have not been properly explored.

MAJOR POINTS:

The manuscript includes a wealth of results and with many of these referred to the supplemental information, I had to continuously look back and forth between the main manuscript and the

supplemental files. I am sure a better balance can be achieved. Additionally, I had a really hard time tracking down what models were being built for what data. Some of it described in the methods but then it was not clear which model was referred to when providing estimates. The given model-derived beta estimates are really hard to interpret – information on effect sizes or something else that would make them meaningful would be great. The use of stable isotopes as sometimes indicating wet habitats (while at the same time having another measure for that) and sometimes as direct stable isotope measures appeared slightly inconsistent adding to the challenging reading.

While I overall find the dataset most interesting and the analyses convincing, I would like to see the differences in dynamics between the dry year and the wetter years more fully explored. From Fig. S1a, it is clear that there was a marked difference between wet and dry years and, that the dynamics also differed between these two situations (larger decrease in wetness in dry years). Given the authors interpretation that birds from poor habitats are “unable” to relocate, it seems surprising that any birds could relocate during the dry year, where conditions even worsened much faster. At the same time there is surprisingly little support for wetness in the proportional hazards models in Belize birds (Table S8), with more for individual “quality” measures (claws) or equal support for condition and wetness (blood). Perhaps that could indicate that individual quality is an important determinant of potential for relocation.

It is a plausible explanation that only birds in a good condition can leave but other potential explanations (predation, density/food availability vs. food abundance) are only very briefly touched upon. I think the authors need to either make it clear why they don’t believe they are important or discuss them more broadly. I would have thought that for example density data are available so that part of this could in fact be quantified.

Overall, I found it challenging to link estimates and models. Given that I have understood the analyses right, I found some of them slightly problematic. What looks like fitted linear models with the explanatory variable on a continuous scale in Fig. 1 does not appear entirely reasonable. They really are data from different years and should be as categorical data. The same issue applies to when year is added as a random factor which it really does not qualify for.

MINOR POINTS (NOT EXHAUSTIVE):

L31/33: “permanent mid-winter movement” – perhaps “permanent mid-winter relocation” would be better.

L158: Why not consider rainfall as a measure of wetness?

L174: Consider providing some more info on stats, sample size, R^2 , model

L185 and L192-195: Not clear how many models were constructed for d15N.

L227: “Error! Reference source not found.” Please correct!

L253: Consider reporting sample size (or df).

L281: Your use of “GPS-tracked population” and “Belize population” confused me for a while. Consider finding a consistent way of referencing that uses either geography or method to separate them.

L281: Note that the two samples are not directly comparable due to the difference in wetness among years in the Belize study.

L286 and 288: Are the two estimates 48.2 km and 124 m correct (one in kilometres the other in meters)?

L303: “regardless of breeding origin”: I couldn’t find this result.

L304-305: “at least partly driven by fluctuating environmental conditions”: Does this refer to a correlation with Coefficient of Variation?

L326-330: I find it odd to have a paragraph that concludes on something that is only shown in the Supplemental Information.

L354: Same as above.

Figure 1: Are these from the general linear models fitted. The variance does not seem to be equal along the axes

Tables in SI: “Null” model not defined

Decision letter (RSPB-2020-2608.R0)

30-Nov-2020

Dear Dr Stanley:

I am writing to inform you that your manuscript RSPB-2020-2608 entitled "Variable tropical moisture and food availability underlie mixed winter space-use strategies in a migratory songbird" has, in its current form, been rejected for publication in Proceedings B.

This action has been taken on the advice of referees, who have recommended that substantial revisions are necessary. With this in mind we would be happy to consider a resubmission, provided the comments of the referees are fully addressed. However please note that this is not a provisional acceptance.

Sincerely,

Dr Maurine Neiman

Associate Editor

Comments to Author:

Two referees have reviewed your manuscript and provide positive evaluations but also request several clarifications. In particular, referee 2 correctly points out that navigating your description of statistical methods is difficult. In particular, it is sometime unclear why year was used as fixed effect in one model but as random effect in another model. On lines 213-214 you state that the reason for including year as a fixed effect in CHP models was that the software currently lacks methods to test model assumptions. It is unclear what you mean by this but it does not seem to be a valid explanation for your assumption. Referee 2 also highlights the absence of effect sizes in the presentation of the results. Relying only on evaluations of fit or significance is not enough. Both referees provide several other very useful comments that you need to take into account.

Reviewer(s)' Comments to Author:

Referee: 1

Comments to the Author(s)

This study successfully combines a wide array of novel approaches to identify environmental correlates driving space-use strategies in a Neotropical migratory bird. The manuscript is well written and provides an interesting contribution to this topic. I have several comments to further improve clarity of the paper.

L 39-40: Keywords should not repeat words from the title

L 51, 59, 346: I suggest to use "movements" instead "migration" here, because intra-tropical "migration" typically differs from directional intercontinental migration by the absence of returning to the starting location.

L 81: You probably mean body condition as you do not use the term "energetic condition" hereafter (see also L 297 and 299).

L 103: "from February-April" should be "from February to April"

L 180-181: Here I suggest to take the time span between the capture and the recapture of each individual into account - the longer span may lead to larger magnitudes of the focal changes and vice versa.

L 227, 278-279, 287: Please check the reference errors.

L 300: I suggest to emphasize the contrast with your expectations in the results section as well.

L 326-328: It is interesting that "females and young birds were more likely to be in lower body condition suggesting behaviourally dominant adult males may exclude subordinate individuals from high-quality sites" but in L 354-356 you state that "neither sex nor age class influenced the probability of departure" and "facultative movements are not driven by competition avoidance by subordinate individuals". However the exclusion of the subordinate females from high-quality sites would lead to movements which, based on your claim, should not be related to sex. Please explain better and discuss briefly.

Fig. 2: Please explain in the figure legend what A and Y on x-axis mean.

Table 1: Please shortly explain how the top models were selected (or refer to the respective section for more details).

Appendix S1: "Ctmm" should be "ctmm" in 4th item in the list of references.

Table S1: The order of supplementary tables should be changed (or missing link to Table S1 added), as you refer to other supplementary tables prior to Table S1.

Referee: 2

Comments to the Author(s)

GENERAL:

This manuscript presents a most interesting data set on the winter strategies of a long-distance migratory songbird. The combination of local-scale and large-scale tracking makes the data ideal for investigating both local individual and larger-scale continental factors. It nicely shows correlations between food and wetness, better condition of birds in wet habitat and it demonstrates surprising higher likelihood of relocation in birds from wetter habitats. However, I found the methods and results section quite challenging to understand and thus, I might easily have misunderstood (and missed) parts. Furthermore, I feel that the difference between dry and wet years and the distinction between condition/bird quality and wetness have not been properly explored.

MAJOR POINTS:

The manuscript includes a wealth of results and with many of these referred to the supplemental information, I had to continuously look back and forth between the main manuscript and the supplemental files. I am sure a better balance can be achieved. Additionally, I had a really hard time tracking down what models were being built for what data. Some of it described in the methods but then it was not clear which model was referred to when providing estimates. The given model-derived beta estimates are really hard to interpret - information on effect sizes or something else that would make them meaningful would be great. The use of stable isotopes as

sometimes indicating wet habitats (while at the same time having another measure for that) and sometimes as direct stable isotope measures appeared slightly inconsistent adding to the challenging reading.

While I overall find the dataset most interesting and the analyses convincing, I would like to see the differences in dynamics between the dry year and the wetter years more fully explored. From Fig. S1a, it is clear that there was a marked difference between wet and dry years and, that the dynamics also differed between these two situations (larger decrease in wetness in dry years). Given the authors interpretation that birds from poor habitats are “unable” to relocate, it seems surprising that any birds could relocate during the dry year, where conditions even worsened much faster. At the same time there is surprisingly little support for wetness in the proportional hazards models in Belize birds (Table S8), with more for individual “quality” measures (claws) or equal support for condition and wetness (blood). Perhaps that could indicate that individual quality is an important determinant of potential for relocation.

It is a plausible explanation that only birds in a good condition can leave but other potential explanations (predation, density/food availability vs. food abundance) are only very briefly touched upon. I think the authors need to either make it clear why they don't believe they are important or discuss them more broadly. I would have thought that for example density data are available so that part of this could in fact be quantified.

Overall, I found it challenging to link estimates and models. Given that I have understood the analyses right, I found some of them slightly problematic. What looks like fitted linear models with the explanatory variable on a continuous scale in Fig. 1 does not appear entirely reasonable. They really are data from different years and should be as categorical data. The same issue applies to when year is added as a random factor which it really does not qualify for.

MINOR POINTS (NOT EXHAUSTIVE):

L31/33: “permanent mid-winter movement” – perhaps “permanent mid-winter relocation” would be better.

L158: Why not consider rainfall as a measure of wetness?

L174: Consider providing some more info on stats, sample size, R^2 , model

L185 and L192-195: Not clear how many models were constructed for d15N.

L227: “Error! Reference source not found.” Please correct!

L253: Consider reporting sample size (or df).

L281: Your use of “GPS-tracked population” and “Belize population” confused me for a while. Consider finding a consistent way of referencing that uses either geography or method to separate them.

L281: Note that the two samples are not directly comparable due to the difference in wetness among years in the Belize study.

L286 and 288: Are the two estimates 48.2 km and 124 m correct (one in kilometres the other in meters)?

L303: “regardless of breeding origin”: I couldn't find this result.

L304-305: “at least partly driven by fluctuating environmental conditions”: Does this refer to a correlation with Coefficient of Variation?

L326-330: I find it odd to have a paragraph that concludes on something that is only shown in the Supplemental Information.

L354: Same as above.

Figure 1: Are these from the general linear models fitted. The variance does not seem to be equal along the axes

Tables in SI: “Null” model not defined

Author's Response to Decision Letter for (RSPB-2020-2608.R0)

See Appendix A.

RSPB-2021-1220.R0

Review form: Reviewer 1

Recommendation

Accept with minor revision (please list in comments)

Scientific importance: Is the manuscript an original and important contribution to its field?

Good

General interest: Is the paper of sufficient general interest?

Good

Quality of the paper: Is the overall quality of the paper suitable?

Good

Is the length of the paper justified?

Yes

Should the paper be seen by a specialist statistical reviewer?

No

Do you have any concerns about statistical analyses in this paper? If so, please specify them explicitly in your report.

No

It is a condition of publication that authors make their supporting data, code and materials available - either as supplementary material or hosted in an external repository. Please rate, if applicable, the supporting data on the following criteria.

Is it accessible?

Yes

Is it clear?

Yes

Is it adequate?

Yes

Do you have any ethical concerns with this paper?

No

Comments to the Author

I like the revision, the manuscript has been considerably improved. I have only several additional minor comments.

L 128, 136, 143: Please define early and late sampling period - are these related with the early and late periods of the dry season (L 163)?

L 228: Please add more details on "variables were standardized following Gelman".

L 235: Please provide standard dates instead the Julian ones.

L 315: 148 should be 148.0 (in line with 1.3).

Table S4: Please add the bottom line.

Decision letter (RSPB-2021-1220.R0)

23-Jun-2021

Dear Dr Stanley

I am pleased to inform you that your manuscript RSPB-2021-1220 entitled "Variable tropical moisture and food availability underlie mixed winter space-use strategies in a migratory songbird" has been accepted for publication in Proceedings B.

The referee(s) have recommended publication, but also suggest some minor revisions to your manuscript. Therefore, I invite you to respond to the referee(s)' comments and revise your manuscript. Because the schedule for publication is very tight, it is a condition of publication that you submit the revised version of your manuscript within 7 days. If you do not think you will be able to meet this date please let us know.

Sincerely,

Dr Maurine Neiman

Associate Editor

Board Member

Comments to Author:

You have satisfactorily addressed all comments made by the reviewers. There are only some minor changes suggested by referee 1, which you need to incorporate. In particular, please explicitly state that the standardisation you use consists in dividing by two standard deviations (which differs from the more common practice of dividing by one sd) as proposed by Gelman (2008). By the way, your reference to this publication contains two instances of 'Stat. Med.' Please remove one of them.

Reviewer(s)' Comments to Author:

Referee: 1

Comments to the Author(s).

I like the revision, the manuscript has been considerably improved. I have only several additional minor comments.

L 128, 136, 143: Please define early and late sampling period - are these related with the early and late periods of the dry season (L 163)?

L 228: Please add more details on "variables were standardized following Gelman".

L 235: Please provide standard dates instead the Julian ones.

L 315: 148 should be 148.0 (in line with 1.3).

Table S4: Please add the bottom line.

Author's Response to Decision Letter for (RSPB-2021-1220.R0)

See Appendix B.

Decision letter (RSPB-2021-1220.R1)

30-Jun-2021

Dear Dr Stanley

I am pleased to inform you that your manuscript entitled "Variable tropical moisture and food availability underlie mixed winter space-use strategies in a migratory songbird" has been accepted for publication in Proceedings B.

Data Accessibility section

Open Access

Paper charges

Sincerely,
Editor, Proceedings B
mailto: proceedingsb@royalsociety.org

Appendix A

Response to Referees: RSPB-2020-2608

*** Denote response to Referees

Reviewer(s)' Comments to Author:

Referee: 1

Comments to the Author(s)

This study successfully combines a wide array of novel approaches to identify environmental correlates driving space-use strategies in a Neotropical migratory bird. The manuscript is well written and provides an interesting contribution to this topic. I have several comments to further improve clarity of the paper.

L 39-40: Keywords should not repeat words from the title

***Thank you, we have corrected it in the manuscript.

L 51, 59, 346: I suggest to use "movements" instead "migration" here, because intra-tropical "migration" typically differs from directional intercontinental migration by the absence of returning to the starting location.

***We have changed intra-tropical migrations to movements.

L 81: You probably mean body condition as you do not use the term "energetic condition" hereafter (see also L 297 and 299).

***Thank you, we have corrected those inconsistencies.

L 103: "from February-April" should be "from February to April"

***Thank you, we have corrected the mistake.

L 180-181: Here I suggest to take the time span between the capture and the recapture of each individual into account - the longer span may lead to larger magnitudes of the focal changes and vice versa.

***We agree with the referee and have added time since capture to the model set examining change in mass.

L 227, 278-279, 287: Please check the reference errors.

***We have corrected all the reference errors and apologize for the confusion.

L 300: I suggest to emphasize the contrast with your expectations in the results section as well.

***We have emphasized the contrast with predictions in the results sections for both study populations.

“Contrary to expectations, individuals had a higher departure probability on wetter plots based on the satellite-derived wetness metric.”

“Similar to radio-tracked wood thrush in Belize and contrary to expectations...”

L 326-328: It is interesting that "females and young birds were more likely to be in lower body condition suggesting behaviourally dominant adult males may exclude subordinate individuals from high-quality sites" but in L 354-356 you state that "neither sex nor age class influenced the probability of departure" and "facultative movements are not driven by competition avoidance by subordinate individuals". However the exclusion of the subordinate females from high-quality sites would lead to movements which, based on your claim, should not be related to sex. Please explain better and discuss briefly.

***We have rephrased this section to be more precise about what was meant by "sites" and "habitats". We were referring more specifically to females and second years being excluded from settling in high quality home ranges within our study plots. Competitive exclusion would therefore result in small scale shifts (local movements) within our study plots when birds are settling and establishing home ranges. The facultative movements we observed during the later part of the winter season are at a much larger scale.

"Here, we did not observe segregation across different habitat types, however, the differences in condition we observed suggest habitat quality for wood thrush may differ across finer spatial scales. Therefore, high quality home ranges may be limited and less competitive birds are behaviourally excluded to drier, sub-optimal home ranges."

Fig. 2: Please explain in the figure legend what A and Y on x-axis mean.

***We have corrected this.

Table 1: Please shortly explain how the top models were selected (or refer to the respective section for more details).

***We have added details on model selection in Table 1 and 2.

"Models were ranked by comparing Akaike's information criteria corrected for small sample (AIC_c). All models with $\Delta AIC_c \leq 2$ were considered equally plausible."

Appendix S1: "Ctmm" should be "ctmm" in 4th item in the list of references.

***We have corrected the reference

Table S1: The order of supplementary tables should be changed (or missing link to Table S1 added), as you refer to other supplementary tables prior to Table S1.

***We agree and the order of the supplementary tables have been corrected

Referee: 2

Comments to the Author(s)

GENERAL:

This manuscript presents a most interesting data set on the winter strategies of a long-distance migratory songbird. The combination of local-scale and large-scale tracking makes the data ideal for investigating both local individual and larger-scale continental factors. It nicely shows correlations between food and wetness, better condition of birds in wet habitat and it demonstrates surprising higher likelihood of relocation in birds from wetter habitats. However, I found the methods and results section quite challenging to understand and thus, I might easily

have misunderstood (and missed) parts. Furthermore, I feel that the difference between dry and wet years and the distinction between condition/bird quality and wetness have not been properly explored.

MAJOR POINTS:

The manuscript includes a wealth of results and with many of these referred to the supplemental information, I had to continuously look back and forth between the main manuscript and the supplemental files. I am sure a better balance can be achieved. Additionally, I had a really hard time tracking down what models were being built for what data. Some of it described in the methods but then it was not clear which model was referred to when providing estimates. The given model-derived beta estimates are really hard to interpret – information on effect sizes or something else that would make them meaningful would be great. The use of stable isotopes as sometimes indicating wet habitats (while at the same time having another measure for that) and sometimes as direct stable isotope measures appeared slightly inconsistent adding to the challenging reading.

***We agree with the referee that the paper requires moving back and forth between the main manuscript and the supplementary information which makes following the results difficult. To improve the flow of the manuscript and make it easier to follow the models presented we have included a summary of the results of the final models (top models or results from model averaging) in table form by adding a new table and editing Table 1 (now Table 2). The tables summarising the AICc model selection remain in the supplementary information to limit the number of tables in the main text. The results obtained from models using blood versus claw isotopes were very similar. Therefore, we have moved most of those results specific to claw isotopes to the supplementary information to further try to simplify the results section. To facilitate interpretation of the model we also tried to clarify in the methods section what data was being used in each model. In the results section, we have tried to be clearer about what models were being discussed.

***To improve the interpretability of our model estimates we have standardized the input variables for all the models. Therefore, the estimates provided now represent a standardized effect size with all factors on a comparable scale. We have clarified this in the Data Analysis section of the results in the main text.

“To facilitate interpretation of the relative strength of parameter estimates, all input variables were standardized following Gelman [46,47]. Therefore, parameter estimates provided represent standardized effect sizes that are on a comparable scale. Parameter estimates with confidence intervals that exclude zero were interpreted as being important predictors in a model”

***To better clarify the two different stable isotopes and how we are using them we have rephrased the relevant sections in the Diet Sampling and Patterns of moisture sections in the methods. In the results section we have also clarified what information each stable isotope is providing (diet vs. habitat moisture) when they are discussed. We think stable isotopes can provide very useful data on fine scale habitat use and diet that would otherwise be difficult to determine in the field. In the methods section we also tried to clarify why we included two measure of moisture. The satellite-derived wetness metric provides an index of moisture at a larger spatial scale (plot-level) compared to stable carbon isotopes which provide an index of

moisture at a finer spatial scale which is relevant to individual birds (based on their habitat and micro-habitat use)

While I overall find the dataset most interesting and the analyses convincing, I would like to see the differences in dynamics between the dry year and the wetter years more fully explored. From Fig. S1a, it is clear that there was a marked difference between wet and dry years and, that the dynamics also differed between these two situations (larger decrease in wetness in dry years). Given the authors interpretation that birds from poor habitats are “unable” to relocate, it seems surprising that any birds could relocate during the dry year, where conditions even worsened much faster. At the same time there is surprisingly little support for wetness in the proportional hazards models in Belize birds (Table S8), with more for individual “quality” measures (claws) or equal support for condition and wetness (blood). Perhaps that could indicate that individual quality is an important determinant of potential for relocation.

*** With the removal of year from the models, the results of our proportional hazard models have changed slightly, however the interpretation remains consistent. When plots are wetter, birds have a higher probability of departure. We have adjusted the discussion to reflect this change. Our results suggest facultative movements are a very common strategy in wood thrush (80% of birds engage in facultative movements). Therefore, even in dry years, such as 2013, the majority of birds leave our site. However, on average site persistence was longer in the drier year. We have added text in the discussion to better illustrate these results.

We also wanted to clarify that the claw measure (d13C) was intended to indicate habitat moisture and not be used as a direct measure of individual quality. We have tried to clarify this further in the main text to avoid confusion. We did find evidence that adult males may be dominant and exclude subordinate (females and young) from wetter home range (where birds were found to be in better condition). However, neither age or sex appeared in the final model explaining site persistence so we don't find any evidence that dominance influenced departure decisions. We also did not find evidence that condition influenced departure probability, however the only measure of condition available was from the early sampling period. Because we only captured birds during two disjoint sampling periods, we were not able to characterise well the condition of birds around the time they departed our site. It would be very interesting to dissect this further in future research.

It is a plausible explanation that only birds in a good condition can leave but other potential explanations (predation, density/food availability vs. food abundance) are only very briefly touched upon. I think the authors need to either make it clear why they don't believe they are important or discuss them more broadly. I would have thought that for example density data are available so that part of this could in fact be quantified.

***Unfortunately, accurate density data is not available on our study sites, but we agree with the referee that it would be very useful data. However anecdotally we don't see a decrease in density across the dry season. As radio-tracked individuals depart, new individuals fill their place. Assessing if density increases would be difficult to determine without standardized surveys. We did not think competition (conspecific or heterospecific) was driving departure decisions because we did not find evidence that subordinate individuals were more likely to depart, as would be expected if competition avoidance was driving these dynamics. However, changes in predator density associated with changes in wetness is a potential explanation and we have added that we

think this topic deserves further exploration. We have elaborated on these additional explanations in the discussion:

“For example, if increased moisture also drives increased movement of conspecifics or heterospecifics (e.g. altitudinal migrants [51]), increased competition for resources could drive movement decisions. However, we did not find support for competitive release by subordinate individuals. Although predation rates were relatively low in our radio-tracked population ($n = 23/181$ individuals), increased density of predators when habitats are wetter could also drive movement decisions and deserves further exploration [54].

Overall, I found it challenging to link estimates and models. Given that I have understood the analyses right, I found some of them slightly problematic. What looks like fitted linear models with the explanatory variable on a continuous scale in Fig. 1 does not appear entirely reasonable. They really are data from different years and should be as categorical data. The same issue applies to when year is added as a random factor which it really does not qualify for.

***We agree with the referee and have removed year as a random factor. With only 3 levels/years of data it seems unlikely that the model will be able to produce precise estimates of the random effect. When we attempted to re-run the models including year as a fixed effect it became apparent that year and our satellite-derived wetness metric were highly collinear. We checked this using the variance inflation factor and a simple ANOVA. To address the collinearity, we have chosen to remove year since collinearity would be expected between year and precipitation (which is strongly linked to the wetness metric).

MINOR POINTS (NOT EXHAUSTIVE):

L31/33: “permanent mid-winter movement” – perhaps “permanent mid-winter relocation” would be better.

***We have adjusted this throughout the manuscript.

L158: Why not consider rainfall as a measure of wetness?

***We agree with the referee that rainfall would be a better variable to measure than wetness. Unfortunately, rainfall measures are not available at a spatial scale that would be relevant to this study. Most satellite derived metrics occur at the scale of 100s of km in Central America. Direct rainfall measurements (e.g. rain gauge) were not available across our study site.

L174: Consider providing some more info on stats, sample size, R^2 , model

***We have made sure to include the estimate (standardized effect size), SE, 95% Cis and sample size for all models either in the text or in Tables 1-2.

L185 and L192-195: Not clear how many models were constructed for $\delta^{15}N$.

***We have added details throughout this paragraph to be more explicit about what data was used in each model.

“To determine the influence of moisture levels on food availability (fruit and insect surveys) and diet ($\delta^{15}N$) we constructed separate models with fruit abundance, insect abundance and $\delta^{15}N$ as response variables.”

Separate models were run for isotopes derived from claws and blood. We clarified this for all models at the end of the Data Analysis section. Results from models run from isotopes derived from blood vs claw largely agreed so we focused the results section on the models constructed based on isotopes derived from blood but included the results obtained from claws in Appendix S1.

“For all models including stable isotopes ($\delta^{15}\text{N}$ or $\delta^{13}\text{C}$), separate models were run for isotopes derived from blood and claws.”

L227: “Error! Reference source not found.” Please correct!

***We apologize again for the errors. They have all been removed and corrected

L253: Consider reporting sample size (or df).

***We have added the degrees of freedom.

L281: Your use of “GPS-tracked population” and “Belize population” confused me for a while. Consider finding a consistent way of referencing that uses either geography or method to separate them.

***We agree with the referee and refer to the two population as the GPS vs radio-tracked populations to avoid confusion.

L281: Note that the two samples are not directly comparable due to the difference in wetness among years in the Belize study.

***We defined the migratory periods separately for the two populations. The start of the migratory period for the radio-tracked populations was defined based on light-level geolocator data collected at our study site over multiple years (earliest known departure). We measured start of the migratory period directly from the GPS-tracked population.

L286 and 288: Are the two estimates 48.2 km and 124 m correct (one in kilometres the other in meters)?

***Yes, the two estimates are correct. We chose to use the different units because we thought it would be easier to interpret.

L303: “regardless of breeding origin”: I couldn’t find this result.

***We have corrected this error. It should read, regardless of winter location.

L304-305: “at least partly driven by fluctuating environmental conditions”: Does this refer to a correlation with Coefficient of Variation?

***We were not specifically referring to the relationship with CV. To avoid confusion, we have adjusted the wording.

“These results provide evidence that space-use strategies in overwintering wood thrush are flexible, and at least partly driven by changes in environmental conditions linked to patterns of moisture.”

L326-330: I find it odd to have a paragraph that concludes on something that is only shown in the Supplemental Information.

***I believe this comment is referring to L333-L338 (now L621-624). The conclusion to this paragraph is referring to the results of our paired t-tests presented in the first paragraph of the results sections (also shown graphically in Appendix S1 Figure S1). We show the changes in environmental condition between our early and late sampling period. Wetness and insect abundance declined, while fruit abundance increased. Given the word count limits we chose to include the graphically representation of the results in the Appendix. Our models analyzing the factors driving food availability (fruit and insect abundance; “*Food availability and diet*” section L238, now L306) also show a relationship between wetness and food availability and is presented in the main text.

L354: Same as above.

***The results discussed in L354 are the results of our Cox proportional hazard models presented in the *Seasonal Site Persistence* section, (L268, now L411) of the main text and Figure 3.

Figure 1: Are these from the general linear models fitted. The variance does not seem to be equal along the axes

***Yes, the line is the predicted relationship and the points represent the raw data. All models were checked for unequal variance using residual plots and outliers were assessed using Cox’s plots.

Tables in SI: “Null” model not defined

***We have defined the null models in the *Model Specifications and Fit Assessment* section of Appendix S1.

“All model sets included an intercept only null model.”

Appendix B

Response to Referees: RSPB-2021-1220

*** Denote response to Referees

Reviewer(s)' Comments to Author:

Referee: 1

Comments to the Author(s).

I like the revision, the manuscript has been considerably improved. I have only several additional minor comments.

L 128, 136, 143: Please define early and late sampling period - are these related with the early and late periods of the dry season (L 163)?

***Thank you, we have clarified what we mean by early and late sampling period in the 3 places mentioned above. The sampling periods are related to the dry season periods. We selected the two sampling periods in order to quantify conditions during the early and late periods of the dry season.

L 228: Please add more details on "variables were standardized following Gelman".

***We have added additional details explaining the standardization procedure. We have also corrected the error in the reference.

“To facilitate interpretation of the relative strength of parameter estimates, all input variables were standardized by subtracting the mean and dividing by two standard deviations following Gelman [46,47].”

L 235: Please provide standard dates instead the Julian ones.

***We have replaced the Julian dates with standard dates.

L 315: 148 should be 148.0 (in line with 1.3).

***Thank you, we have corrected the inconsistency.

Table S4: Please add the bottom line.

***Thank you, we have added the bottom line.